# Risk Stratification after an Acute Coronary Syndrome: Significance of Antithrombotic Therapy

**DOI:** 10.3390/jcm10081572

**Published:** 2021-04-08

**Authors:** Victoria A. Brazhnik, Larisa O. Minushkina, Olga I. Boeva, Niyaz R. Khasanov, Elena D. Kosmacheva, Marina A. Chichkova, Dmitry A. Zateyshchikov

**Affiliations:** 1Central State Medical Academy of Department of Presidential Affairs, 121359 Moscow, Russia; vabrazhnik@mail.ru (V.A.B.); minushkina@mail.ru (L.O.M.); dz@bk.ru (D.A.Z.); 2City Clinical Hospital No. 51 of the Moscow City Healthcare Department, 121309 Moscow, Russia; 3Department of Propedeutics of Internal Diseases, Kazan State Medical University, Ministry of Health Care of the Russian Federation, 420012 Kazan, Russia; ybzp@mail.ru; 4Department of Propedeutics of Internal Diseases, Kuban State Medical University, Ministry of Health Care of the Russian Federation, 350063 Krasnodar, Russia; kosmachova_h@mail.ru; 5Astrakhan State Medical University, Ministry of Health Care of the Russian Federation, 414000 Astrakhan, Russia; m.chichkova@mail.ru

**Keywords:** acute coronary syndrome, risk predictors, antithrombotic therapy, de-escalation

## Abstract

The impact of the de-escalation strategy of antiplatelet therapy (APT) on the life expectancy after acute coronary syndromes (ACS) and percutaneous coronary intervention (PCI) requires an assessment in real clinical practice. Into the Russian multicentral observational trial (ORACLE II ClinicalTrials.gov number, NCT04068909), 1803 patients with ACS and PCI indications were enrolled. During 12 months of follow-up, 228 all-cause deaths have occurred. The analysis of death predictors was carried out by the classification tree method. Age, an option of antithrombotic therapy, a history of chronic heart failure, and uric acid level had the greatest prognostic value. The death prediction model’s sensitivity was 82.1% in the training cohort and 79.2% in the test cohort. During the observation period, ticagrelor was replaced with clopidogrel (APT de-escalation) in 357 patients. The groups of patients with different antiplatelet therapy options were adjusted for clinical parameters by the pseudorandomization method. The de-escalation group had the lowerest all-cause death rate. The incidence of bleeding and recurrent nonfatal coronary events in the study groups did not differ significantly. Thus, the APT regimen’s advantage of changing from the maximum in the first weeks after ACS to moderate at follow-up has been confirmed. There is an obvious need to study the possibilities of individualizing antiplatelet therapy in patients after acute coronary syndromes.

## 1. Introduction

The management of patients with the acute coronary syndrome (ACS) involves long-term dual antiplatelet therapy (APT), including acetylsalicylic acid (ASA) and a P2Y_12_ inhibitor. In current clinical guidelines on the management of acute coronary syndromes among P2Y_12_ inhibitors, the priority is given to more modern and more active drugs—ticagrelor and prasugrel [1,2]. However, in real clinical practice, only some of patients takes these drugs. They are often replaced by clopidogrel (de-escalation of APT) due to therapy’s side effects, the risk of bleeding, and financial reasons. The 2018 European Society of Cardiology guidelines on coronary revascularization include a provision that a de-escalation strategy under platelet aggregation control may be considered after percutaneous coronary interventions (PCI) [3]. This recommendation is based on the results of the TROPICAL-ACS (Testing Responsiveness to Platelet Inhibition on Chronic Antiplatelet treatment for Acute Coronary Syndromes) trial, where replacement of prasugrel for clopidogrel 2 weeks after hospital discharge was comparable in the risk of adverse events if the more active antiaggregant was continued [4]. However, the implementation of a new approach also requires assessing the impact on the life expectancy of patients who have undergone ACS in real practice. The new approaches to data processing based on machine learning methods seem promising. One of them is building classification and regression trees, which can be used in the processing of large datasets from studies of real practice.

The present study aimed to analyze the risk predictors of adverse outcomes and evaluate the efficacy and safety of different APT options in patients after ACS in real clinical practice.

## 2. Materials and Methods

The present data was obtained from the Russian multicentral observational trial—ORACLE II (ObseRvation after Acute Coronary syndrome for deveLopment of trEatment options; reg. number: clinicaltrials.gov NCT04068909). ORACLE II was performed in four invasive cardiology hospitals in Moscow, Kazan, Krasnodar, and Astrakhan in 2014–2017. The study protocol of ORACLE II was described in detail previously [5]. A total of 1803 patients (1120 (62.1%) men, average age 64.9 ± 12.78 years) with ACS and indications for PCI at the index hospitalization were enrolled in the study. Of these patients, 682 (37.8%) had an ACS with ST-segment elevation, 1584 (87.9%) suffered from arterial hypertension (AH), 410 (22.7%) from diabetes mellitus, and 311 (17.2%) from atrial fibrillation. A total of 529 (29.3%) patients had a history of myocardial infarction, 213 (11.8%) of cerebral stroke, and 907 (50.3%) patients had signs of chronic heart failure (CHF). A PCI was given to 1013 (56.2%) patients during the index hospitalization.

All subjects gave their informed consent for inclusion before they participated in the study. The study was conducted in accordance with the Declaration of Helsinki, and the protocol was approved by the Ethics Committee of the Central State Medical Academy of the Presidential Executive Office Directorate (№ 14/14 from 20 October 2014).

Study personnel carried out prescheduled visits or telephone contacts on days 25, 90, 180, and 360 after hospital discharge, collecting drug adherence and endpoints information by questionnaire. By the study’s observational nature, all treatment decisions were left to the patient’s care team’s discretion. All-cause deaths, cardiovascular deaths (coronary events, pulmonary embolism, and chronic heart failure), recurrent nonfatal coronary events (ACS and unplanned revascularization), and strokes were recorded as adverse outcomes. All bleeding cases were also considered, and their severity was assessed according to BARC (Bleeding Academic Research Consortium) classification [6]. During the visits, data on drug therapy were registered, including names and dosage of medications and the patient’s own assessment of regularity of intake.

Statistical processing of the results was performed with the SPSS 23.0, MedCalc 19.0.3, and Minitab 19 software. An analysis of the distribution and criteria for its normality was performed using the Shapiro–Wilks method for extended variables. Since the distribution of all studied variables was normal, parametric methods were used for analysis. Mean values and errors (M ± m) were calculated for extended variables. Discrete variables were compared by Pearson’s χ^2^ criteria. Differences were considered statistically significant if a *p*-value was less than 0.05.

Sensitivity analysis did not show any difference between patients from various study centers.

All-cause mortality predictors were analyzed by CART (Classification and Regression Tree) algorhythm to create a decision tree. Validation methods for predictive analytics techniques were k-fold cross-validation (training and test cohorts as 1:5). The number of patients in the test cohort was 429.

We used ROC (receiver operating characteristic) analysis with an assessment of the AUC (area under ROC curve) indicator to evaluate the quality of the predictive model. The model’s quality was rated as excellent at an AUC value of 0.9–1.0; 0.8–0.9—very good, 0.7–0.8—good, 0.6–0.7—average, and 0.5–0.6—unsatisfactory.

The groups of patients with different APTs were adjusted using a pseudorandomization method, using a propensity score calculated by logistic regression. The groups were adjusted for 1:1 nearest-neighbor search by age, frequency of PCI, renal function, and the presence of aortic stenosis. Diabetes mellitus and blood hemoglobin levels at admission were considered as covariates before inclusion. 

Survival was analyzed by the Kaplan–Meier method using the log-rank test.

## 3. Results

During the observation period, 228 all-cause deaths were registered. The distribution of timing and pattern of causes of death are shown in Table 1.

As possible clinical predictors, we considered patients’ clinical characteristics, laboratory testing data, echocardiography parameters, patient status at hospital discharge, antithrombotic and hypolipidemic therapy—189 predictors in total. The final model included 39 significant predictors of death. An optimal classification model, including 14 terminal nodes, was developed. The minimum number of patients in a terminal node was five. The probabilities of events for each node are shown in Figure 1.

The relative prognostic significance of predictors is presented in Figure 2. Age, the antithrombotic therapy option after hospital discharge, a history of CHF, and uric acid levels were the most significant predictors of all-cause mortality.

The final model data were validated on the test cohort. The sensitivity of the model was 82.1% in the training cohort and 79.2% in test one. The area under the ROC curves in training and test cohorts did not practically differ (Figure 3).

The analysis has resulted in identifying the group with high risk (503 patients) in the initial cohort. The high-risk group’s mortality rate was 31.6%, while in the group with low risk, it was 5.1% (*p* < 0.0001).

We analyzed the two most significant factors in the classification tree—the nature of antithrombotic therapy at the outpatient stage and the patient’s age—in more detail.

After the first episode of ACS, 1652 patients were discharged from the hospital. At the time of discharge, 134 patients were receiving dual APT or triple APT with oral anticoagulants. These patients were not included in this analysis. The ATP treatment of the rest of 1518 patients at discharge and at last known follow-up visit is shown in Table 2.

For those who changed therapy from ticagrelor to clopidogrel at different stages of follow-up (357 patients), this change was registered on the 25th day (168 patients), on the 90th day (91 patients), and on the 180th day (98 patients).

Table 3 shows the clinical characteristics of the patients receiving ticagrelor or clopidogrel as part of dual APT for the entire follow-up period and the patients with de-escalation of APT.

As shown in Table 3, patients who received clopidogrel as a part of APT were significantly older. More often, they had concomitant diabetes mellitus, aortic stenosis, and renal dysfunction. There was a lower proportion of those with ACS with ST-segment elevation among these patients; they were less likely to receive PCI during the index hospitalization. The patients receiving ticagrelor throughout the follow-up and those in the de-escalation group were similar in their clinical characteristics. However, the de-escalation group had a lower proportion of patients who received PCI on initial admission, and a lower hemoglobin level was noted. The de-escalation group had the lowest number of all-cause deaths as well as coronary deaths. The highest proportion of deaths was among patients receiving clopidogrel. Most bleeding was registered on ticagrelor therapy, but the major bleeding incidence did not differ significantly between the groups.

Table 4 shows mean follow-up time and mean survival time; survival probability depending on the APT regimen is shown in Figure 4. Maximum time to the endpoint “all-cause death” was registered in the de-escalation group, and the minimum time was registered in the group of patients treated with the combination of ASA and clopidogrel. Survivors’ follow-up time did not differ in the studied groups (Table 4). 

The differences in survival function were reliable (Figure 4).

The present survival probability pattern was common in patients over the age of 67 years, as well as in younger patients (Figure 5 and Figure 6).

A pseudorandomization procedure was performed to account for the significant differences in the groups with different APT regimens (Table 3). We adjusted the groups for age, frequency of PCI, presence of aortic stenosis, and renal function. Diabetes mellitus and blood hemoglobin levels at admission were considered as covariates. Due to pseudorandomization, we formed groups with comparable major clinical characteristics (Table 5).

As can be seen from Table 5, after the adjustment, the differences in the incidence of all-cause mortality and coronary death between patients prescribed ticagrelor as a part of APT and patients after de-escalation of APT disappeared. The incidence of death was higher in patients receiving clopidogrel. The incidence of bleeding and recurrent nonfatal coronary events did not differ significantly between the groups under study. Figure 7 shows the survival curves in the groups.

## 4. Discussion

The actual rate of APT de-escalation and substitution of ticagrelor and prasugrel with clopidogrel is quite high. In the Swiss study in a cohort of 1278 patients undergoing PCI, for 17% of outpatients, ticagrelor was discontinued earlier than recommended. Most commonly (57% of cases), ticagrelor was replaced with clopidogrel (de-escalation); for 28% of patients, ticagrelor was withdrawn, and for 16% of patients, it was replaced with prasugrel. The most frequent reason for discontinuing ticagrelor was bleeding, less frequently it was side effects, such as dyspnea and breathing difficulties [7]. In the Swedish National ACS Registry, about 10% of patients initially prescribed ticagrelor or prasugrel for ACS switched to clopidogrel within 1 year [8]. The type of basic drug may influence the de-escalation rate. The TRANSLATE-ACS (Treatment with Adenosine Diphosphate Receptor Inhibitors: Longitudinal Assessment of Treatment Patterns and Events after Acute Coronary Syndrome) study, which enrolled 12,365 ACS patients after PCI, demonstrated the switching to clopidogrel occurred in 28.3% of ACS patients discharged on ticagrelor-based therapy compared with 15.4% discharged on prasugrel-based one [9]. In the PRAGUE-18 (Prasugrel Versus Ticagrelor in Patients With Acute Myocardial Infarction Treated With Primary Percutaneous Coronary Intervention)observational study, de-escalation of therapy and change of P2Y_12_ inhibitor to clopidogrel was also more frequent in patients initially receiving ticagrelor (44%) compared with receiving prasugrel (34%). De-escalation of therapy was often carried out for economic reasons; simultaneously, the number of ischemic events did not increase but even turned out to be less than on treatment with ticagrelor and prasugrel [10]. Prasugrel was not approved in Russia at the time of the study. Additionally, some studies demonstrate the lower prevalence of switching more potent antiplatelet agents to clopidogrel. The Spanish multicenter observational study showed only a 2% de-escalation in 1717 ACS patients [11]. In our study, the de-escalation rate of APT at the outpatient stage was higher. Switching to clopidogrel occurred more frequently at the patient’s visits of the 25th and 90th days after hospital discharge. The reasons for the de-escalation of therapy were not analyzed.

The main problem regarding de-escalation of therapy remains whether such a treatment regimen is sufficiently effective and does not lead to an increase in the number of ischemic events. In a Singapore study of patients with ACS with ST-segment elevation, who were observed in 2014–2016, of 349 patients initially receiving ticagrelor as a part of dual APT, 219 patients changed ticagrelor to clopidogrel after hospital discharge. The risk of recurrent atherothrombotic events did not differ significantly during the follow-up, while the risk of clinically significant bleeding was slightly lower with de-escalation (7.8% vs. 8.5%; *p* = 0.047) [12]. Of 4678 ACS patients initially receiving ticagrelor in the Chinese registry, 1019 had replaced it with clopidogrel during the one year of follow-up. In 380 patients, such replacement was done earlier after ACS (first 30 days), in 639 it was done later. The late replacement was most often associated with bleeding. Simultaneously, the incidence of atherothrombotic events in the patients who switched therapy later was lower than among patients who received ticagrelor during the year. The incidence of major bleedings did not differ significantly between these two groups [13].

In the randomized TOPIC (timing of platelet inhibition after acute coronary syndrome) trial, the patients 1 month after ACS were randomized to continue therapy with ticagrelor or prasugrel or de-escalate antithrombotic therapy. The incidence of atherothrombotic events did not differ significantly, and major bleeding was less common in the de-escalation group [14].

A meta-analysis of 13 studies analyzing data from 17,896 patients undergoing PCI was published in 2020. Deescalation of APT was reported in 4105 (23%) of the enrolled patients. De-escalation of therapy was slightly more frequent earlier (1 month) after PCI. The number of all atherothrombotic events, as well as the number of coronary deaths, recurrent myocardial infarctions, strokes, and major bleedings, did not differ significantly between patients receiving standard APT and those after de-escalation [15].

In our study, the rate of all-cause mortality and coronary deaths was lower in the de-escalation group than in standard therapy patients. It should be noted that the differences persisted after the groups were adjusted for the main clinical characteristics. The incidence of all recurrent coronary events did not differ significantly. At the same time, the differences in clinical characteristics were initially very significant. After the adjustment procedure, the number of patients in each group turned out to be small, which was a limitation of the study. 

Somewhat unexpected results were revealed in the bleeding analysis. The frequency of bleeding did not differ significantly between the groups after the pseudorandomization procedure. Thus, the hypothesis about the association of the differences in mortality with bleeding frequency was not confirmed in our study. Similar data were obtained in one of the real clinical practice studies. Namely, bleeding rates were higher in the de-escalation group than in primarily dual APT with clopidogrel group and lower than in patients receiving ticagrelor and prasugrel during the year. However, the differences in bleeding rates leveled off after the pseudorandomization procedure [16]. These differences, possibly, can be explained by microbleeding into the atheroma, which may underlie aseptic inflammation and the development of the atherothrombotic cascade. Limitations of the study: The sample size turned out to be insufficient for carrying out the propensity search procedure for all different parameters without a significant loss in the number of groups. Additionally, the study protocol did not include registration of the adverse drug effects options, which did not make it possible to analyze the reasons for changing an-tithrombotic therapy during follow-up fully. This analysis also did not include low-ther-apy-adherent patients.

## 5. Conclusions

Thus, the advantage of changing the APT regimen from a maximum in the first weeks after ischemic heart disease exacerbation to moderate at follow-up has been confirmed. Based on CART algorhythm results, we can speculate that the dependence of de-escalation significance of age may be a key point in developing an individualized treatment regimen. Our work shows the need for special studies of the individualization of antiplatelet therapy in this category of patients. 

## Figures and Tables

**Figure 1 jcm-10-01572-f001:**
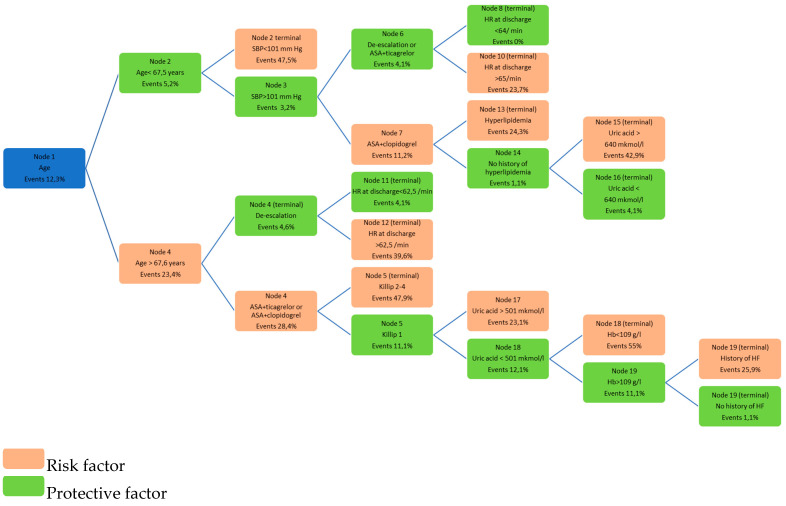
Probabilities of the event development for each terminal node in the optimal classification model. Footnote: HR-heart rate, ASA-acetylsalicylic acid, HF-heart failure, Hb-hemoglobin, SBP-systolic bood pressure.

**Figure 2 jcm-10-01572-f002:**
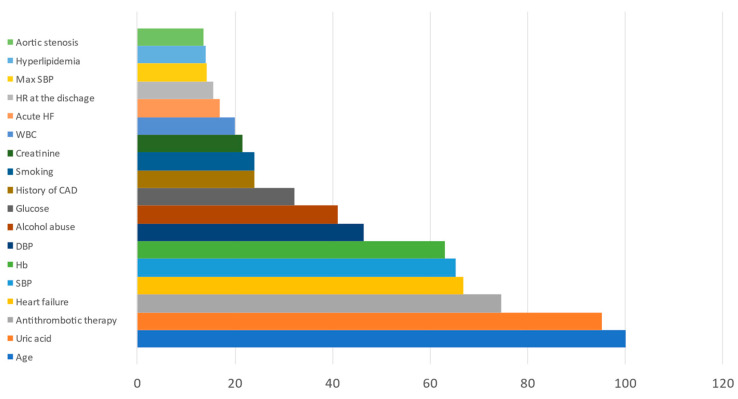
Relative prognostic significance of clinical predictors of all-cause mortality.

**Figure 3 jcm-10-01572-f003:**
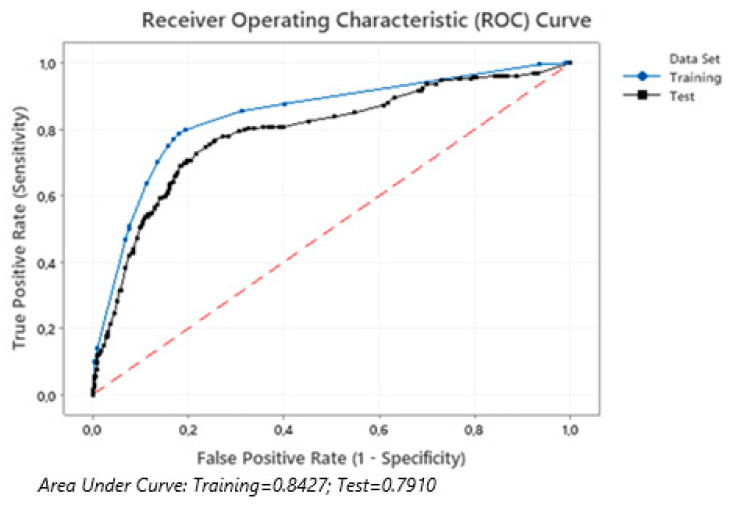
The area under the ROC curves of the all-cause mortality prediction model for patients in the training and test cohorts.

**Figure 4 jcm-10-01572-f004:**
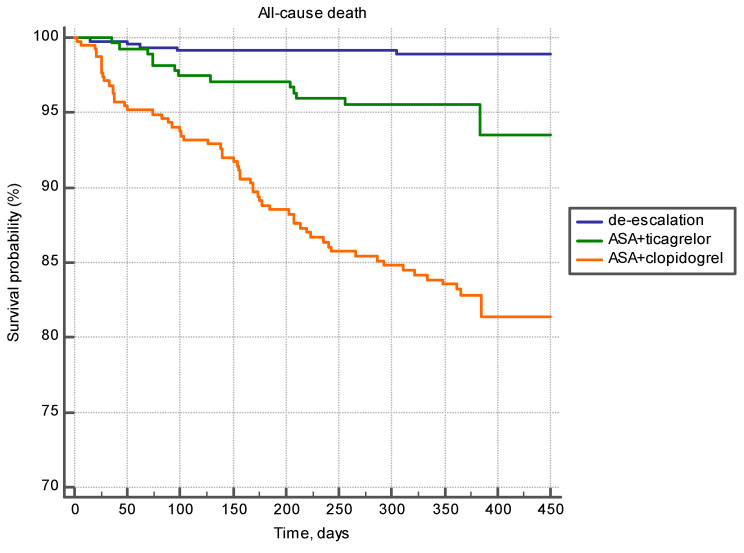
Survival probability at different regimens of APT (Kaplan–Meier analysis).

**Figure 5 jcm-10-01572-f005:**
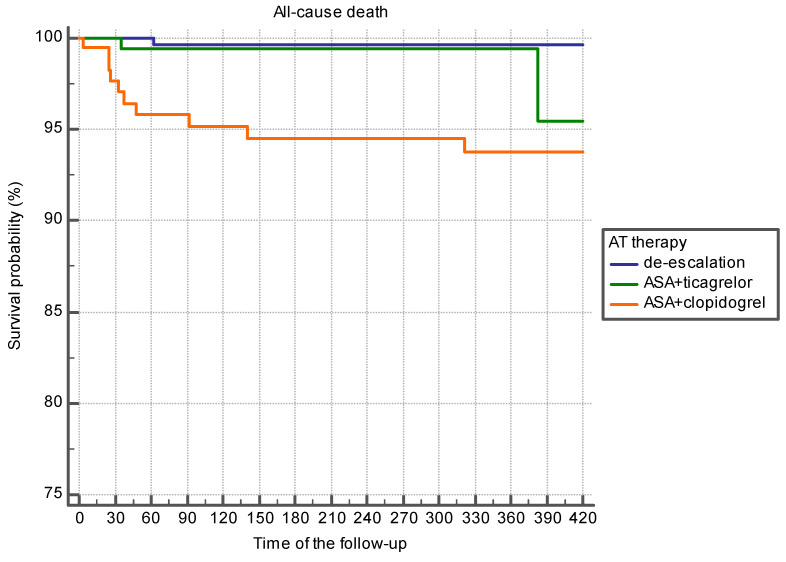
Survival probability of patients under 67 years depending on the APT options (Kaplan–Meier analysis).

**Figure 6 jcm-10-01572-f006:**
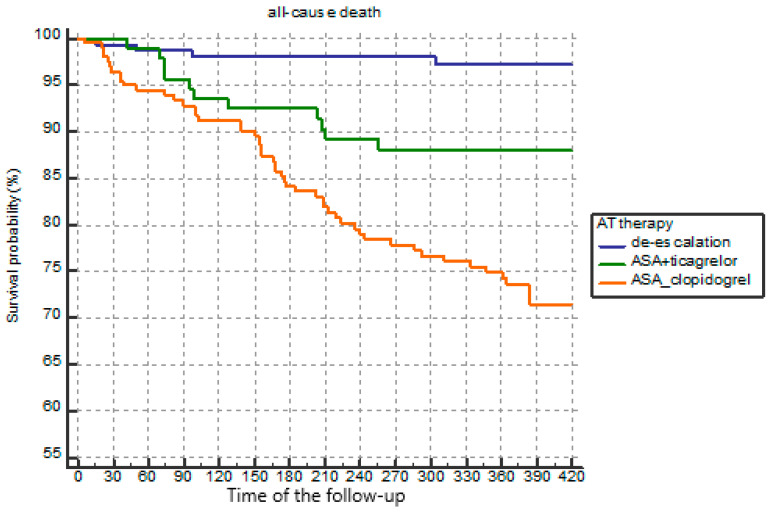
Survival probability of patients over 67 years depending on the APT options (Kaplan–Meier analysis).

**Figure 7 jcm-10-01572-f007:**
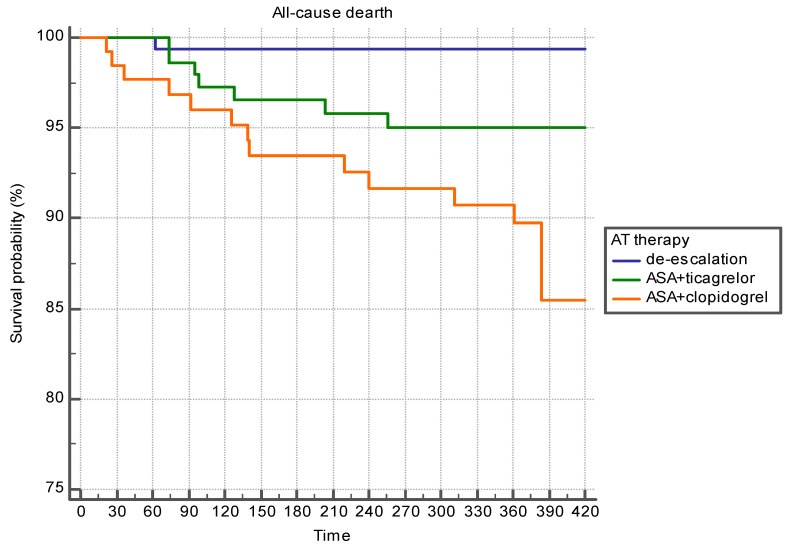
Survival probability depending on APT options after pseudorandomization (Kaplan–Meier analysis).

**Table 1 jcm-10-01572-t001:** Distribution of causes of death by the time of follow-up.

Visit/Reason	Inpatient Care (Primary Hospital Admission)	25 Days	90 Days	180 Days	360 Days
All-cause mortality, including:	104	30	29	30	35
- coronary	90	19	9	17	9
- stroke	2	2	4	1	4
- pulmonary embolism (PE)	2	1	1	2	
- chronic heart failure (CHF)	4	5	5	5	6
- other	6	–	7	5	12
- unspecified	–	3	3	–	4

**Table 2 jcm-10-01572-t002:** Antiplatelet therapy (APT) at hospital discharge and the last known follow-up visit.

Therapy	At Discharge	At Last Known Follow-Up Visit	Under 67.5 Years of Age	Over 67.5 Years of Age
Without APT	12	92	39	53
Monotherapy (ASA or clopidogrel)	127	254	103	151
ASA + clopidogrel	739	434	210	224
ASA + ticagrelor	640	283	175	108
De-escalation (changing from ticagrelor to clopidogrel)	–	357	193	164
No data or indication of different therapy at all visits (low adherence)	–	98	31	67

**Table 3 jcm-10-01572-t003:** Clinical characteristics of the patients on different regimens of APT.

Variable	ASA + Ticagrelor (*n* = 283)	ASA + Clopidogrel (*n* = 434)	De-Escalation (*n* = 357)	*p*
Age in years, (M ± m)	61.83 ± 12.419	65.58 ± 12.550	61.82 ± 11.816	0.001
Gender, male, *n* (%)	188 (66.4%)	255 (58.8%)	245 (68.5%)	0.040
ACS with ST-segment elevation, *n* (%)	120 (42.4%)	138 (31.8%)	164 (46.0%)	0.005
Increased markers of myocardial damage at the index event, *n* (%)	277 (97.8%)	360 (83.0%)	336 (94.1%)	0.001
Killip II–IV at the index event, *n* (%)	51 (18.1%)	74 (17.0%)	91 (25.6%)	0.012
History of ischemic heart disease, *n* (%)	191 (67.4%)	302 (69.6%)	255 (71.4%)	0.556
Arterial hypertension, *n* (%)	244 (86.1%)	382 (88.0%)	308 (86.3%)	0.756
Type II diabetes mellitus, *n* (%)	62 (21.9%)	122 (28.2%)	70 (19.5%)	0.035
Heart failure, *n* (%)	118 (41.7%)	198 (45.6%)	171 (47.9%)	0.182
Aortic stenosis, *n* (%)	13 (4.6%)	42 (9.6%)	13 (3.7%)	0.002
Peripheral atherosclerosis, *n* (%)	52 (18.4%)	97 (22.4%)	95 (26.5%)	0.088
Gastrointestinal erosive and ulcerative lesions, *n* (%)	35 (12.4%)	61 (14.0%)	36 (10.1%)	0.476
History of gastrointestinal bleeding, *n* (%)	16 (5.7%)	23 (5.2%)	16 (4.5%)	0.883
Hemoglobin on admission, g/l	139.64 ± 17.839	133.54 ± 21.685	134.60 ± 18.249	0.043
Creatinine on admission, µmol/l	97.65 ± 27.916	101.98 ± 51.571	97.59 ± 27.458	0.088
Glomerular filtration rate (GFR), ml/min	82.64 ± 27.850	76.66 ± 33.845	82.11 ± 37.537	0.001
GFR < 60 mL/min/1.73 m^2^	60 (21.1%)	158 (36.4%)	81 (22.8%)	0.001
PCI during hospitalization, *n* (%)	237 (83.7%)	181 (41.8%)	215 (60.2%)	0.0001
Adverse outcomes, *n* (%)				
All-cause death, *n* (%)	15 (5.3%)	34 (7.8%)	5 (1.4%)	0.0001
Coronary death, *n* (%)	5 (1.8%)	42 (9.7%)	2 (0.6%)	0.001
Recurrent coronary events	43 (15.1%)	62 (14.2%)	41 (11.6%)	0.354
Bleeding of any kind, *n* (%)	54 (19.1%)	27(6.2%)	34 (9.4%)	0.001
Clinically significant bleeding (2–5 BARC), *n* (%)	15 (5.3%)	15 (3.5%)	9 (2.6%)	0.144
Major bleeding (3–5 BARC), *n* (%)	2 (0.7%)	10 (2.3%)	4 (1.1%)	0.283

Footnote: FC-functional class, PCI-percutaneous coronary intervention.

**Table 4 jcm-10-01572-t004:** The time before all-cause death at different regimens of APT (Kaplan–Meier analysis).

Variable	Mean Follow-Up Time (before All-Cause Death or Last Known Visit)	*p*	Mean Expected Survival Time by Kaplan–Meier Analysis	*p*	Mean Follow-Up Time, Excluding Deaths	
Observation data in real practice						
De-escalation	372.4 ± 4.47	<0.0001	911.1 ± 3.97	<0.0001	375.4 ± 4.31	0.878
ASA + ticagrelor	368.4 ± 7.88		757.7 ± 18.254		378.3 ± 7.56	
ASA+ clopidogrel	292.3 ± 9.3737		706.9 ± 18.72		373.8 ± 10.12	
Groups formed as a result of pseudorandomization						
De-escalation	377.2 ± 8.08	<0.0001	846.8 ± 5.15	0.001	379.2 ± 7.86	0.921
ASA + ticagrelor	368.4 ± 12.003		784.1 ± 12.54		379.9 ± 11.69	
ASA + clopidogrel	303.6 ± 15.95		752.1 ± 26.77		376.7 ± 16.76	

**Table 5 jcm-10-01572-t005:** Clinical characteristics of the groups formed as a result of pseudorandomization.

Variable	ASA + Ticagrelor (*n* = 149)	ASA + Clopidogrel (*n* = 149)	De-Escalation (*n* = 149)	*p*
Age in years, (M ± m)	63.83 ± 10.64	64.51 ± 11.051	64.32 ± 11.049	0.898
Gender, male, *n* (%)	91(61.1%)	87 (58.4%)	83 (55.7%)	0.643
ACS with ST-segment elevation, *n* (%)	48 (32.2%)	46 (30.9%)	51 (34.2%)	0.824
Increased markers of myocardial damage at the index event, *n* (%)	114 (76.5%)	114 (76.5%)	115 (77.2%)	0.988
Killip II–IV at an index event, *n* (%)	26 (17.4%)	17 (11.4%)	24 (16.1%)	0.309
History of ischemic heart disease, *n* (%)	112 (75.2%)	111 (74.5%)	106 (71.1%)	0.700
Arterial hypertension, *n* (%)	131 (87.9%)	131 (87.9%)	130 (84.4%)	0.980
Type II diabetes mellitus, *n* (%)	34 (22.8%)	35 (23.5%)	29 (19.5%)	0.667
Heart failure, *n* (%)	66 (44.3%)	73 (49.0%)	76 (51.0%)	0.493
Aortic stenosis, *n* (%)	2 (1.3%)	2 (1.3%)	2 (1.3%)	0.999
Peripheral atherosclerosis, *n* (%)	28 (18.8%)	28 (18.8%)	26 (17.4%)	0.988
Gastrointestinal erosive and ulcerative lesions, *n* (%)	21 (14.1%)	20 (13.4%)	22 (14.7%)	0.884
History of gastrointestinal bleeding, *n* (%)	10 (6.7%)	12 (8.1%)	12 (8.1%)	0.886
Hemoglobin on admission, g/l	138.6 ± 17.21	139.7 ± 16.87	138.81 ± 18.863	0.567
Creatinine on admission, µmol/l	97.12 ± 22.782	99.26 ± 21.421	98.01 ± 17.185	0.788
Glomerular filtration rate (GFR), ml/min	81.15 ± 31.031	77.62 ± 25.076	80.12 ± 27.454	0.564
GFR < 60 mL/min/1.73 m^2^	38 (25.5%)	37 (24.4%)	39 (26.2%)	0.886
PCI during hospitalization, *n* (%)	115 (77.2%)	115 (77.2%)	115 (77.2%)	0.999
Adverse outcomes, *n* (%)				
All-cause death, *n* (%)	7 (4.7%)	13 (8.7%)	2 (1.3%) *^,¥^	0.013
Coronary death, *n* (%)	3 (2.0%)	7 (4.7%)	1 (0.7%) **^,χ^	0.074
Recurrent nonfatal coronary events, *n* (%)	22 (14.8%)	20 (13.4%)	22 (14.8%)	0.930
Bleeding of any kind, *n* (%)	23 (15.4%)	22(14.8%)	18 (11.7%)	0.679
Clinically significant bleeding (2–5 BARC), *n* (%)	8 (5.4%)	11(7.4%)	7 (4.5%)	0.589
Major bleeding (3–5 BARC), *n* (%)	1 (0.7%)	4 (2.7%)	1 (0.7%)	0.219

* *p* = 0.003 compared to the ASA + clopidogrel group; ^¥^
*p* = 0.086 compared to the ASA + ticagrelor group; ** *p* = 0.023 compared to the ASA + clopidogrel group; ^χ^
*p* = 0.302 compared to the ASA + ticagrelor group.

## Data Availability

The data presented in this study are available on request from the corresponding author. The data are not publicly available due to regulatory restrictions.

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
