# Peer review of "Risk Stratification after an Acute Coronary Syndrome: Significance of Antithrombotic Therapy"

_jcm, 2021, doi:10.3390/jcm10081572_

Round 1
Reviewer 1 Report
Brazhnik et al present a original clinical research article entitled "Risk Stratification after an Acute Coronary Syndrome: Significance of Antithrombotic Therapy". They measured ihe impact of the de-escalation strategy of antiplatelet therapy (APT) on the life expectancy after acute coronary syndromes (ACS) and percutaneous coronary intervention (PCI). 1803 patients with ACS and PCI indications were enrolled. During 12 months of follow-up, 228 all-cause deaths occurred. They show that in their study the de-escalation group had the lowerest all-cause death rate. The incidence of bleeding and recurrent nonfatal coronary events in the study groups did not differ significantly. They conclude taht the APT regimen's advantage of changing from the maximum in the first weeks after ACS to moderate at follow-up has been confirmed.
Question on the cohorts : the study was done in cohorts in various part of Russia (Moscow, Kazan, Krasnodar, and Astrakhan). Did the authors acount for ethnic (and thus genetci ) differences?
Where the patients followed-up with clinical (in-person) follow-up visits in a dedicated outpatient clinic or scheduled for phone interviews? This can induce a significant difference in follow-up. See PMID 31433760
A part of the discussion seems to be missing. It starts with "and prasugrel with clopidogrel is quite high. In"
In Conclusions ,the sentence "This section is not mandatory but can be added to the manuscript if the discussion is unusually long or complex." should not appear.
There are only 14 referenes which is too few. Authors coudl for exemple more discuss their reults in the context of their renal failure patients;
A recapitulative figure would be welcome.
line 245 please rephrase"De-escalation of APT was registered in 4105 (23%) patients underwent."
Author Response
Please, see the attachment.

Reviewer 2 Report
The authors used CART classification and created a prognostic model which could accurately identify the predictors with the greatest prognostic value for adverse outcomes. Importantly, using propensity-score matching, the authors concluded that the incidence of death was significantly lower among ACS patients, who followed a de-escalation dual antiplatelet strategy after the first weeks of the index event.
The manuscript is well-written and the study concept is very interesting. The authors employ robust statistical analyses, which confirms the internal validity of their findings. My only concern is that study limitations are not adequately described in the discussion.
Author Response
Please, see the attachment.

Round 2
Reviewer 1 Report
changes are ok